# A New Kv1.3 Channel Blocker from the Venom of the Ant *Tetramorium bicarinatum*

**DOI:** 10.3390/toxins17080379

**Published:** 2025-07-30

**Authors:** Guillaume Boy, Laurence Jouvensal, Nathan Téné, Jean-Luc Carayon, Elsa Bonnafé, Françoise Paquet, Michel Treilhou, Karine Loth, Arnaud Billet

**Affiliations:** 1Equipe BTSB EA-7417, Institut National Universitaire Champollion, Place de Verdun, 81012 Albi, France; guillaume.boy@univ-jfc.fr (G.B.); nathan.tene@univ-jfc.fr (N.T.); jean-luc.carayon@univ-jfc.fr (J.-L.C.); elsa.bonnafe@univ-jfc.fr (E.B.); michel.treilhou@univ-jfc.fr (M.T.); 2Centre de Biophysique Moléculaire, Centre National de la Recherche Scientifique (CNRS), Unité Propre de Recherche (UPR) 4301, 45071 Orléans, France; laurence.jouvensal@cnrs-orleans.fr (L.J.); francoise.paquet@cnrs-orleans.fr (F.P.); karine.loth@cnrs-orleans.fr (K.L.); 3Unité de Formation et de Recherche (UFR) Sciences et Techniques, Université d’Orléans, 45071 Orléans, France

**Keywords:** ant venom peptide, Kv channels, functional dyad

## Abstract

Ant venoms are rich sources of bioactive molecules, including peptide toxins with potent and selective activity on ion channels, which makes them valuable for pharmacological research and therapeutic development. Voltage-dependent potassium (Kv) channels, critical for regulating cellular excitability or cell cycle progression control, are targeted by a diverse array of venom-derived peptides. This study focuses on MYRTX_A4_-Tb11a, a peptide from *Tetramorium bicarinatum* venom, which was previously shown to have a strong paralytic effect on dipteran species without cytotoxicity on insect cells. In the present study, we show that Tb11a exhibited no or low cytotoxicity toward mammalian cells either, even at high concentrations, while electrophysiological studies revealed a blockade of hKv1.3 activity. Additionally, Ta11a, an analog of Tb11a from the ant *Tetramorium africanum*, demonstrated similar Kv1.3 inhibitory properties. Structural analysis supports that the peptide acts on Kv1.3 channels through the functional dyad Y21-K25 and that the disulfide bridge is essential for biological activity, as reduction seems to disrupt the peptide conformation and impair the dyad. These findings highlight the importance of three-dimensional structure in channel modulation and establish Tb11a and Ta11a as promising Kv1.3 inhibitors. Future research should investigate their selectivity across additional ion channels and employ structure-function studies to further enhance their pharmacological potential.

## 1. Introduction

Animal venoms are complex chemical cocktails composed of numerous molecules whose primary function is to kill and/or paralyze prey [1]. The venoms of many species are extensively studied in human health, not only to develop antivenoms but also to identify new pharmacological agents for research or new therapeutic and agronomic molecules. In the search for new drugs in human medicine, special attention has been paid to peptide toxins, which offer numerous advantages. In particular, their three-dimensional structures often provide exceptional specificity and selectivity for pharmacological targets at low active concentrations [2]. In particular, venom peptides exhibit a wide range of biological effects and target diverse families of ion channels, many of which have been explored as pharmacological agents or therapeutic compounds [3].

Voltage-dependent potassium (Kv) channels are no exception and are the targets of numerous peptide toxins from diverse species [4]. Structurally, Kv channels are tetrameric ion channels, with each subunit consisting of six transmembrane helices (S1–S6). The voltage-sensing domain (S1–S4) detects changes in membrane potential, while the pore-forming domain (S5–S6) houses the selectivity filter on the extracellular side of the membrane. This is a highly conserved region made up of a five-residue sequence (TVGYG), termed the signature sequence, within each subunit, that is critical for potassium ion specificity and channel blockade by toxins [5,6].

Potassium channel-inhibiting toxins represent a diverse and highly specialized group of peptides. Their inhibitory activity is often linked to their mode of interaction with the potassium channel, which can be broadly categorized into two mechanisms: gate modulation and pore blockade [7].

In the gate modulating mechanism, toxins do not directly occlude the pore but instead alter the gating properties of the channel, affecting its activation or inactivation states. These toxins can be classified into voltage-sensor trapping toxins and allosteric modulators, depending on their mode of action.

In the pore blockade mechanism, toxins physically obstruct the potassium channel pore, preventing K^+^ conduction. While it is not common to all pore blocking peptides, it has been shown that their activity can rely on a charged/aromatic residue pair defined as a functional dyad [8]. Typically, a conserved lysine (or arginine) residue inserts into the pore, directly blocking ion flow, while an adjacent aromatic residue (e.g., tyrosine, phenylalanine) stabilizes binding through interactions with the selectivity filter and/or outer-vestibule residues. For example, the potent Kv blocker ShK from sea anemone features a K22-Y23 dyad that demonstrates this mechanism [9]. However, some toxins, despite possessing this dyad, behave almost as if they did not, relying instead on multi-site stabilization to block the channel [10,11]. Others lack a functional dyad altogether but still achieve effective pore blockade through alternative interaction networks [11].

Although not extensively studied so far, ant venoms represent a promising new field of exploration for the discovery of bioactive molecules. Ants, with more than 16,000 species identified to date, of which 70% are venomous, possess venoms that are primarily composed of peptides [12]. These venoms serve multiple functions such as predation, defense or communication, and contribute to ants’ adaptation to various ecological niches. Using a proteo-transcriptomic approach, the venom composition of numerous ant species has been characterized and shows remarkable molecular diversity, including peptide repertoires with distinct biological activities [13]. For example, the venom of the ant *Tetramorium bicarinatum* comprises a complete peptide repertoire of 37 peptides, categorized into 17 toxin groups based on sequence similarity [14]. Among these, MYRTX_A1_-Tb0a and MYRTX_A1_-Tb9a exhibit cytotoxic activity by targeting cell membranes [15], while MYRTX_A2_-Tb3a acts as a Nav modulator, inducing pain in vertebrates [16]. Another peptide, P17 (MYRTX_A1_-Tb1a), functions as an agonist of the GPCR MGRPX2, triggering pro-inflammatory responses [17].

Recently, MYRTX_A4_-Tb11a (Tb11a), derived from *T. bicarinatum* venom, has been identified as a non-cytotoxic peptide with strong paralytic activity in insects such as flies and honeybees [18]. Additionally, results from pharmacological tests conducted on S2 drosophila cell line suggest that Tb11a modulates potassium conductance. Moreover, Tb11a exhibits a compact three-dimensional helical structure, stabilized by a disulfide bond between residues C10 and C33, and features a potential functional dyad consisting of K25 and Y21, superposable to the dyads observed in Kv channel-targeting toxins such as ShK, BgK, and ChTx. This led us to hypothesize that Tb11a might exert its paralytic effect by targeting Kv channels. Surprisingly, in a recent article, Peigneur et al. found that Tb11a did not modulate any of the many Kv channels they examined, including Kv1.3, but had a strong cytotoxic activity in oocyte model [19]. As these results contradict our own findings, we carefully examined the Tb11a peptides tested in both studies. Peigneur et al. mistakenly used the measured monoisotopic mass of the reduced form to conclude that the peptide was oxidized, with the disulfide bridge formed. Thus, the disulphide bridge appears to be crucial for structural stabilization and, consequently, for biological activity.

In this paper, we report the effect of the fully structured and disulfide-bonded Tb11a peptide on mammalian cells. We assessed the cytotoxicity of the peptide on various cell lines and observed no cell death at low active concentration, even though a slight cytotoxicity was found at high concentrations. Structural elements guided us to investigate the modulation of voltage-dependent potassium channels by Tb11a, with a focus on Kv1.3 channel (KCNA3) and the voltage- and Ca^2+^-activated K^+^ channel KCa1.1 (slo1). Electrophysiological recordings showed a blockade of Kv1.3 channel activity and a slight but significant inhibition of KCa1.1 channels. We also evaluated the activity of the *T. africanum* homolog peptide (Ta11a) and found similar results on Kv1.3. Docking experiments between either of the peptides, and Kv1.3 confirms the importance of the functional dyad.

## 2. Results

### 2.1. Tb11a Exhibits No Cytotoxic Activity in HEK293T

Venoms are used by ants for predation and defense and often contain numerous cytotoxic compounds [15,20,21]. We have previously demonstrated that Tb11a is not cytotoxic to insect cells, which are closely related to the ecological target of ant venoms [18]. However, data on its effect on mammalian cells is lacking. To address this, we evaluated the cytotoxicity of the Tb11a peptide and its effect on cell viability after a 24 h incubation on HEK293T cell line.

Tb11a cytotoxicity was assessed at concentrations between 1 and 50 µM and showed no cytotoxicity at lower concentrations (≤10 µM) and low cytotoxic activity at 50 µM, measured at 10.35% mortality after 24 h of exposure (Figure 1). Cell viability was also unaffected by low concentrations of Tb11a, even though a slight but statistically significant decrease in cell viability was observed at 10 and 50 µM (12.19% and 10.23%, respectively). These results show that Tb11a does not affect cell viability nor mortality at low concentrations, including those used for electrophysiological experiments (Figure 2). Cytotoxic activities were also studied on various cell lines such as epithelial breast cells (the cancerous cell line MCF-7 and the non-cancerous control MCF-10A) and neuronal cells (SH-SY5Y cells), with no effect on cell viability and mortality (Appendix A). On the contrary, the reduced Tb11a induces a cell mortality detectable at concentrations as low as 1 µM (Appendix A).

### 2.2. Tb11a Is a Kv1.3 Channel Blocker

Preliminary experiments showing a Na^+^-independent modulation of the KCl-induced membrane depolarisation of *Drosophila* S2 cells exposed to Tb11a [18], as well as the presence of a functional dyad in Tb11a that is found in many voltage-gated potassium channel blockers, suggest that Tb11a may interact with Kv channels. To test this hypothesis, KCa1.1 and Kv1.3 human potassium channels were heterologously expressed in HEK293T cells, and the patch clamp technique (voltage clamp, whole-cell configuration) was used to assess the effect of Tb11a on channel activity (Figure 2 and Appendix A). As shown in Figure 2, when HEK293T were successfully transfected with hKv1.3, bath application of 1 µM Tb11a induced a 43.37 ± 5.44% decrease in the peak potassium current amplitude without altering current kinetics. To ensure that the elicited current originated from the target channel, non-transfected HEK293T and GFP-transfected HEK293T cells were used as negative controls, and 4-AP, a well-known Kv1.3 inhibitor, was used as a positive control. To further investigate the activity of Tb11a, its effect on KCa1.1 channel—also considered a promising candidate based on preliminary data and the peptide biophysical properties—was also tested. This time, paxilline, a mycotoxin known to be a BK channel blocker, was used as a positive control. Interestingly, the current elicited by hSlo1-transfected cells was also affected upon addition of Tb11a, albeit to a lesser extent, displaying 24.75 ± 6.13% decrease in peak current at +80 mV (Appendix A). These results suggest that Tb11a is not specific to Kv1.3 and could also affect other subtypes of potassium channels. Interestingly, application of the reduced Tb11a did not modulate the Kv1.3 current, highlighting the importance of the disulfide bridge in the biological activity of the peptide (Figure 3D).

### 2.3. Ta11a, an Analog of Tb11a also Affects Kv1.3

Tb11a displays an original structure never reported for an ant venom peptide, presenting two putative functional dyads (residues K2-F11 and Y21-K25) consisting of a stabilizing hydrophobic amino acid residue and a basic lysine that can occlude the selectivity filter of potassium channels. Interestingly, Ta11a (previously labeled U11-MYRTX-Ta1a), an analog of Tb11a sharing 73.5% of sequence identity in the mature region, was found in the ant *Tetramorium africanum* and displays only one of the potential functional dyads found in Tb11a. To test this hypothesis, human Kv1.3 was transiently expressed in HEK293T, and the effect of Ta11a was assessed using patch clamp technique (Figure 3). Ta11a addition caused a decrease in voltage-elicited currents similar to that observed upon Tb11a addition, with 40% of inhibition. These results suggest that potassium-channel inhibitory properties of both peptides may rely on the shared dyad, which is composed of Y21 and K25 and could play the pharmacophore role in both peptides.

### 2.4. Structure Analysis of Ta11a

The 3D structure of Ta11a was determined by homology based on the NMR structures of Tb11a. The structures of the two analogs are shown in Figure 4. As expected, they are highly similar and share the same compactness. Interestingly, Ta11a possesses an additional, very short, 3.10-helix (E13-L16) compared to Tb11a, where D13-L16 forms a type IV β-turn. In both peptides, the α-helix interacts with the last 3.10-helix, forming an inter-helix angle of −120.7 ± 2.3° in Tb11a and −106.3 ± 0.7° in Ta11a. The flexible type IV β-turn in Tb11a contrasts with the more rigid 3.10-helix in Ta11a and probably influences the hydrophobic core stability and the inter-helix angle.

All charged residue side chains are exposed to the solvent, contributing to the electrostatic properties and solubility of peptide. Given the peptide’s inherent tendency to form helices, K25 adopts a precise orientation at the vertex of the triangular ring helix structure, and this position is uniquely constrained by the disulfide bridge and the compact loop that connects the α- and 3.10-helices. This particular orientation of K25 could be critical for its role in the functional dyad.

### 2.5. Models of the Tb11a-Kv1.3 and Ta11a-Kv1.3 Complexes

Indeed, although electrophysiological data suggest that Tb11a and Ta11a act as Kv1.3 blockers, the exact mechanism remains unclear. We hypothesize that their activity involves the interaction of the Y21-K25 functional dyad with the channel. To explore this, we used Haddock 2.4 to build three-dimensional models of the complexes. Instead of imposing unambiguous distance restraints, Ambiguous Interaction Restraints (AIRs) were applied to guide the docking, focusing on potential interactions between K25 and residues in the narrowest part of the pore region named the selectivity filter (T444-G448 of the four subunits), and in the outer vestibule (D449-H451 of the four subunits) of the potassium channel.

The analysis of 200 water-refined conformations of Tb11a-Kv1.3 and Ta11a-Kv1.3 complexes resulted in a subset of models where K25 interacted with the selectivity filter, specifically forming hydrogen bonds with Y447. The presence of a hydrogen bond was assessed by measuring the distance between the nitrogen (N) of the N_ζ_H^3+^ side chain group of K25 and the backbone carbonyl oxygen (O) of Y447 for the four channel subunits. Subsequent selection prioritized structural compactness of the toxin, resulting in a subset of high-confidence models with well-preserved features and favorable HADDOCK scores. A comparative summary of the conformation selection process is presented in Table 1.

As a result, the structure of Tb11a in the complex was analyzed in the 42 selected models (see Appendix A), revealing a consistently compact conformation comprising up to four helices: an α-helix (L7-C10, present in 86% of the models), a 3.10-helix (D13-T15, present in 55% of the models), a second α-helix (A18-H24, present in 74% of the models), and another 3.10-helix (E27-H29, present in 69% of the models).

Similarly, the structure of Ta11a in complex with Kv1.3, analyzed across the 45 selected models (see Appendix A), featured three prevalent helices: a 3.10-helix (E13-L16, present in 91% of the models), a α-helix (A18-K23, present in 91% of the models), and another 3.10-helix (E27-H29, present in 51% of the models).

### 2.6. Toxin-Kv1.3 Interfaces

The planar triangle of the Tb11a peptide appears to lie at the entrance to the channel pore. Tb11a engages all four Kv1.3 subunits through a combination of hydrogen bonds, hydrophobic interactions, and a salt bridge, ensuring robust binding (see Figure 5). At the core of this interaction are the functional dyad residues K25 and Y21, which play distinct roles. K25 forms critical hydrogen bonds with the conserved Y447 on all four Kv1.3 subunits (A, B, C, and D), anchoring the toxin at the selectivity filter, while Y21 contributes to stability through hydrophobic interactions with two opposing subunits (B and C). The N-terminal region, including D13 and L16, makes contact with residues of the pore turret P424, T425, and S426 on subunit B, anchoring the initial position of toxin. Further along, the central region, particularly H24, interacts with G448 and H451 on subunit C and D449 on subunit D, bridging the outer vestibule and selectivity filter. At the C-terminal end, E27 forms a stabilizing salt bridge with H451 on subunit A. Thus, despite a non-palindromic sequence, Tb11a establishes a symmetrical interaction network reflecting the fourfold symmetry of Kv1.3 (see Figure 7), which probably contributes to enhance its binding stability and pore-blocking efficacy.

Like Tb11a, Ta11a binds all four Kv1.3 subunits using a combination of hydrogen bonds and hydrophobic interactions. The functional dyad residue K25 anchors the toxin to the selectivity filter by forming hydrogen bonds with Y447 on all four subunits (A, B, C, and D) with 100% prevalence (see Figure 6). The second dyad residue, Y21, enhances stability through hydrophobic and hydrogen bonding interactions with G448 and D449 on subunit B, which are observed in 97.8% of the models. Beyond the dyad, H24 plays an important role by interacting with G448 and D449 on both subunits C and D, as well as with H451 on subunit C, strengthening the connection to the outer vestibule. At the C-terminal end, I26 provides additional, albeit less frequent, stabilization through contacts with D449 and H451 on subunit D. Thus, despite sequence differences, Ta11a resembles Tb11a in establishing a symmetrical interaction network across the Kv1.3 four subunits (see Figure 7).

## 3. Discussion

Despite the richness of their composition, ant venoms contain only few peptides with complex three-dimensional structures, like those of the many ion channel modulators reported in the literature [13]. Even though several biological effects have already been reported for ant venom peptides, including sodium channel inhibitory activities [16], no potassium channel modulator had been discovered so far. In this study, we have identified a new Kv1.3 channel inhibitor, Tb11a, from the venom of the ant *T. bicarinatum*. Tb11a also proved to be a KCa1.1 channel inhibitor, but to a lesser extent. Moreover, we have shown that its analog Ta11a, from the venom of the ant *T. africanum*, shares its inhibitory properties on Kv1.3.

Our research group previously demonstrated that Tb11a induced reversible paralysis in several insect species, suggesting modulation of nerve transmission independent of neurotoxicity [18]. This hypothesis was further supported by studies on KCl-elicited membrane potential variations in insect cell lines, which we demonstrated to be independent of Na^+^ flux. Finally, structural analysis led us to focus on voltage-gated potassium channels, with a view to a more in-depth study of the mode of action of Tb11a. Indeed, Tb11a features a triangular and almost planar ring closed by a disulfide bridge. In addition, Tb11a also exhibits one of the key features of voltage-gated potassium channel blocking peptides, namely the presence of a conserved functional dyad, which plays a critical role in Kv1.x, KCa1.1 and KCa3.1 channel inhibition [22]. This dyad typically consists of a positively charged lysine residue, which occludes the pore by engaging in electrostatic interactions with the channel’s selectivity filter [23], and an adjacent aromatic residue, often a tyrosine or phenylalanine, that enhances binding through hydrophobic and cation-π interactions. The sidechains of both residues are located within ~7Å of each other. The precise positioning of these residues within the peptide structure is essential for high-affinity and selective binding to Kv channels pore, making the dyad a hallmark of Kv channel blockers found in various venomous species. This common feature is notably found in structurally unrelated potassium channel-blocking peptides from scorpion or sea anemone venoms [9]. Tb11a and its analog, Ta11a, share the functional dyad Y21-K25.

As expected, the structures of Tb11a and Ta11a are highly similar (Figure 4), reflecting the high sequence identity between the two toxins and the fact that the structure of Ta11a was derived through homology modeling. The structural integrity of the two peptides is critically dependent on the C10-C33 disulfide bridge, which plays a pivotal role in maintaining their compact, triangular conformation. This bridge stabilizes the fold of the peptide by bringing together the α-helix and the second 3.10-helix, thereby constraining their respective orientations. Indeed, without the disulfide bridge, structural models predict significant conformational changes, with the peptide either adopting a single long α-helical conformation—an unstable structure in aqueous environments without detergents or membrane-mimicking systems—or becoming highly flexible, with only residues 17–24 retaining an α-helical fold. The bridge may thus be essential for preserving the biologically active structure of the peptide, by facilitating the formation of a hydrophobic core, limiting structural flexibility, and ensuring a proper spatial arrangement of key residues. As a matter of fact, the precise orientation of K25 at the vertex of the triangular ring structure, driven by both the disulfide bond and the peptide’s inherent tendency to form helices, is critical for its role in the functional dyad. Together, these features ensure that the functional dyad can be optimally positioned for an effective channel blockade.

In this respect, the results of the docking experiments provide valuable structural insights.

The docking for Ta11a revealed a binding interface that mirrors the interaction network of Tb11a with Kv1.3, confirming that both toxins may employ a conserved mechanism to block the channel. Docking results, while not directly reflective of ligand affinity and subject to limitations, such as the use of a homology model for Ta11a or a greater number of AIRs for the Ta11a adjusted docking, provide valuable structural insights (Appendix A).

The observed similarities between the Ta11a and Tb11a docking results suggest a largely conserved binding strategy, relying on the functional dyad. K25 forms conserved hydrogen bonds with Y447 on all four Kv1.3 subunits, anchoring the toxins symmetrically at the pore. Y21 interacts with G448, D449, and H451 in Tb11a, and these interactions are reproduced in the Ta11a models, particularly with G448 and D449, emphasizing the conserved nature of these key contacts.

Beyond the dyad, other residues contribute to stabilizing interactions with Kv1.3 (Appendix A). In Tb11a, H24 bridges the outer vestibule and selectivity filter, forming strong interactions with G448, D449, and H451, a pattern also observed in the Ta11a models, albeit with slightly reduced frequencies. Additionally, Tb11a benefits from a unique salt bridge between E27 (Tb11a) and H451 (Kv1.3), providing further stabilization to the complex, while this specific salt bridge is absent in the Ta11a-Kv1.3 models. The non-conserved position 451 (Tyr for Kv1.1, Val for Kv1.2, His for Kv1.3) has been identified as critically defining the affinity of a number of blocking toxins for Kv1 channel subtypes [24]. These findings suggest that both Tb11a and Ta11a toxins rely on a conserved interaction network centered on the dyad and supported by secondary stabilizing residues, with subtle differences in their peripheral interactions reflecting minor adaptations in binding strategies. In particular, residues D13, L16, A17 and H29 of Tb11a interact with pore turret residues. In contrast to the selectivity filter, this region is the most variable of the receptor. Residues in this variable region have been shown to be implicated in the selectivity of scorpion α-type potassium channel blockers toward Kv1.x and KCa1.1 channels [25].

In addition to its significant inhibition of Kv1.3, Tb11a inhibit KCa1.1 to a lesser extent. Such features were observed with the potassium channel blockers ShK and BgK. Both peptides were able to decrease Kv1.3 and KCa3.1, another channel from the KCa family. The affinity was far greater for Kv1.3 than for KCa3.1 (1000- and 4-fold for ShK and BgK, respectively) [26].

Structural differences in the pore-turret and outer-vestibule regions surrounding the conserved selectivity filter may explain the differing inhibition capacities of Tb11a on both channels. A key substitution at the position equivalent to H451 in Kv1.3, replaced by tyrosine (Y) in hSlo1, presumably alters the chemical environment of the vestibule. Unlike histidine, which can participate in salt bridges when protonated, tyrosine lacks the ability to form such ionic interactions due to its neutral phenolic side chain. This substitution likely eliminates stabilizing interactions critical for anchoring the toxins. Other substitutions, such as V453 (Kv1.3) to K (hSlo1), T425 (Kv1.3) to L (hSlo1), and S426 (Kv1.3) to T (hSlo1), further modify the topology and chemistry of the vestibule, precluding effective binding of the toxins.

As previously mentioned, Peigneur et al. recently published a paper investigating the effect of Tb11a on various types of ion channels, and did not observe any inhibitory activity, including on Kv1.3 channel. As these results contradict our own findings, we carefully examined the Tb11a peptides tested in each study. Mass spectrometry results suggest that Peigneur et al. worked with the reduced form of Tb11a (monoisotopic mass of 4020.24, expressed as [M+H^+^]), without the disulfide bridge that stabilizes the triangular ring helix. As a reminder, the monoisotopic masses of the two oxidized peptides are 4018.14 Da (4019.14 as [M+H^+^]) for Tb11a and 4017.11 Da (4018.11 as [M+H^+^]) for Ta11a (LC-ESI-MS analysis of oxidized and reduced Tb11a are shown Appendix A). We therefore decided to test the cytotoxicity of the reduced Tb11a as well as its activity on Kv1.3 currents. A strong cytotoxicity was observed on HEK cells and no inhibition of Kv1.3 current at the tested concentrations (Figure 3 and Appendix A) confirming the data obtained by Peigneur et al., [19] and the importance of the disulfide bond in the biological activity. Taken together, these results further strengthen the functional dyad hypothesis. Moreover, in addition to impairing peptide structure, thereby negating the peptide binding to the channel, the reduction in Tb11a could also confer cytotoxic properties by redistributing charges, since it is a highly charged peptide. Linearized Tb11a could interact with phospholipidic membranes, as is well known for the venom of many arthropods, including ants [15]. Linear helical peptides are well known for exhibiting cytotoxic activity by forming pores in cell membranes [15], which is in accordance with the observation by Peigneur et al., that oocytes become leaky upon application of concentrations of reduced Tb11a higher than 1 μM. Based on genomic data and a clustering analysis of signal sequences, venom peptides precursors of *Myrmicinea* have been classified into 8 families [27,28]. Tb11a and Ta11a belong to the A4 superfamily together with Tb21a, Ta21a and b and seven other peptides found in tribes other than *Tetramorium* (representative sequences of superfamily A4 peptides from *Tetramorium*, *Manica* and *Myrmyca* ants are presented in Appendix A). In particular, we found that *tb11a* and *tb21a* genes are located next to each other on chromosome 4 [28]. Interestingly, except for Tb11a and Ta11a, all members of the A4 family are all linear and amphipatic peptides predicted to be helicoidal. These biochemical properties are common to cytotoxic peptides. While cytotoxic activity has to date never being studied for A4 linear peptides, Mri20a has been reported to be insecticidal with a PD50 of 70 nmol.g^−1^ on bowflies [29]. The absence of disulphide bonded peptides in the A4 family outside the *Tetramorium* genus may indicate that this type of peptide arose specifically in *Tetramorium*, probably from an A4 ancestral gene encoding linear peptide.

## 4. Conclusions

Taken together, our results showed that Tb11a and its analog Ta11a were able to significantly decrease voltage elicited Kv1.3 currents. This study, combined with our previous work [18] and the study of Peigneur et al. [19] strongly suggest that these properties mainly rely on a functional dyad Y21-K25.

Given the fact that Kv1.3 dysregulation is associated with multiple pathological conditions, the need for specific modulators remains current. As Kv1.3 plays a significant part in regulating the membrane potential and Ca^2+^ concentration in T cells, it has been linked to various autoimmune diseases such as rheumatoid arthritis, multiple sclerosis, psoriasis, and type-1 diabetes [30]. In addition, interest in Kv1.3 as an oncogenic target is growing. The channel contributes to several carcinogenesis stages like cell proliferation, survival, apoptosis and immune escapes mechanisms in multiple types of cancer [31]. Therefore, further characterization of the properties of Tb11a and Ta11a will be needed. Their selectivity must be assessed on more ion channels, particularly potassium channels. Structure-function studies could also reveal deeper details about the binding mechanism of the peptide and could improve the effectiveness and the selectivity of both peptides.

## 5. Materials and Methods

### 5.1. Cell Culture

HEK293 cells (generously gifted by Pr. Frederic Becq, Poitiers, French) and MDA-MB-231 cells were cultured in high glucose DMEM supplemented with 10% fetal bovine serum (FBS) and 1% penicillin/streptomycin. MCF10-A (ATCC) were cultured in high glucose DMEM supplemented with 5% horse serum, 1% penicillin/streptomycin, 500 µg/mL hydrocortisone, 10 mg/L insulin and 200 ng/mL epithelial growth factor (EGF). All cell lines were grown in standard conditions (37 °C, 5% CO_2_).

### 5.2. Transient Transfection of Potassium Channels

pEGFP-hKv1.3 plasmid (pEGFP-C1 vector) was kindly provided by Pr. Zoltan Varga (University of Debrecen, Hungary), pcDNA-hSlo1 and pcDNAhBeta1 plasmids were kindly provided by Pr. Chris Lingle (Washington University, St Louis, MO, USA). In case of KCa1.1, pcDNA vectors were co-transfected with a pCMV-GFP [32] (gift from Connie Cepko; Addgene plasmid #11153) in a ratio of 3:1 (channel plasmid: GFP). GFP-positive transfectants were identified with an Olympus fluorescence IX73 microscope (Olympus, Tokyo, Japan) and were used for current recordings. For patch clamp experiments, HEK293 cells were transfected with the corresponding cDNA construct using cationic lipids (JetPEI; QBiogene, Illkirch, France) with 0.2 µg/mL plasmid.

### 5.3. Cytotoxicity Assays

CCK-8 (BosterBio, Pleasanton, CA, USA) and LDH (Dojindo, Rockville, MD, USA) assay kits were used to determine cell viability and mortality, respectively.

Briefly, cells were plated at a density of 5000 cells per well in a 96-well plate and incubated overnight at 37 °C. Cells were then exposed to various peptide concentrations ranging from 1 to 50 µM for 24 h before assay in accordance with the manufacturer’s instructions, as previously described [15].

### 5.4. Patch Clamp Experiments

Ionic currents were measured in the whole cell configuration of the patch clamp method. Voltage-clamp signals were recorded using an analog/digital interface (Digidata 1440; Axon Instruments, Inc., Burlingame, CA, USA) and analyzed using pCLAMP version 10 (Axon Instruments) (List Electronic, Darmstadt, Germany). The holding potential was −80 mV and current/voltage (*I*/*V*) relationships were built by clamping the membrane potential to −80 mV and by pulses from −80 to 80 mV in 10 mV increments. Pipettes were prepared by pulling borosilicate glass capillary tubes (GC150-TF10; Clark Electromedical Inc., Reading, UK) using a two-step vertical puller (Narishige, Tokyo, Japan). Pipette capacitance was electronically compensated in cell-attached mode. The external bath solution contained (in mM): 141 NaCl, 4.7 KCl, 1.2 MgCl_2_, 1.8 CaCl_2_, 10 glucose, and 10 HEPES (titrated with NaOH to pH 7.4). The osmolarity was 300 mOsmol. The intrapipette solution contained (in mM): 125 KCl, 4 MgCl_2_, 10 HEPES, 10 EGTA, 5 MgATP (titrated with KOH to pH 7.2). The osmolarity was 290 mOsmol. All experiments were conducted at room temperature (20–25 °C). The calculated chemical equilibrium potential for potassium (*E*_K_+) is −82.5 mV.

### 5.5. Chemicals and Peptide Synthesis

All peptides were chemically synthetized by GenScript (Rijswijk, Netherlands) with 95% purity. Tb11a disulfide bridge was reduced by incubated oxidized Tb11a with 100 mM ammonium bicarbonate buffer (pH 8) containing 10 mM dithiothreitol (DTT) for 30 min at 56 °C. The reduction in the disulfide bridge was controlled by LC-ESI-MS analysis (Appendix A).

All other products were from Sigma-Aldrich (Saint-Quentin-Fallavier, France). Stock solutions of peptides (2.2 mM) and 4-aminopyridine (4-AP) (100 mM) were solubilized in ultrapure water and stored at −40 °C until use. Stock solution of paxilline (2 mM) was solubilized in dimethylsulfoxide (DMSO) and stored at −20 °C until use.

### 5.6. Statistics

Data are expressed as the means ± SEM of *n* observations. Data normality was assessed with the Shapiro–Wilk’s test and compared using Student’s test or one way ANOVA followed by Dunnett’s test with GraphPad Prism version 8.0.2 (GraphPad Software). Differences were considered significant when *p* < 0.05.

### 5.7. Homology Modeling of Ta11a

Five models of the Ta11a toxin were generated through the Robetta webserver with the comparative modeling method [33] using the Tb11a NMR structures (PDB 8PWT) as templates [18].

### 5.8. Docking of Tb11a and Ta11a to Kv1.3

Docking to determine the binding interactions between Tb11a or Ta11a and Kv1.3 was conducted using HADDOCK version 2.4 [34]. The docking analysis required three-dimensional structures of both partners. The cryo-EM structure of a Fab-Shk/Kv1.3 complex (PDB ID: 7SSV) as used for Kv1.3 [35], while the NMR structure of Tb11a (PDB ID: 8PWT) and homology models of Ta11a were used for the peptides. Following the approach outlined by Sanches et al., [36] docking was focused on the pore-vestibule region of the four subunits of Kv1.3. Residues T443 to D449 were defined as active residues, and residues T441 to H451 were set as fully flexible. Passive residues were automatically selected by HADDOCK. For Tb11a and Ta11a, residue K25 was specified as the only active residue, with passive residues automatically assigned by HADDOCK. The entire peptide was modeled as fully flexible during the docking process.

From the 200 water-refined structures generated for each complex, we analyzed and selected models based on the following criteria:The presence of multiple potential hydrogen bonds between the N_ζ_H_3_^+^ side-chain ammonium group of K25 and the backbone carbonyl oxygen (O) of Y447 across the four subunits of Kv1.3 (distance (N_ζ_-O) < 4 Å).Structural compactness and the number of secondary structure elements in the peptides, as the disulfide bridge is known to confer reduced flexibility and enhanced fold stability in a tridimensional structure.The best HADDOCK scores among the structures selected according to criteria (1) and (2).

The selected structures were analyzed using PDBSum [37]. All figures were performed using PyMOL version 3.1 [38].

## Figures and Tables

**Figure 1 toxins-17-00379-f001:**
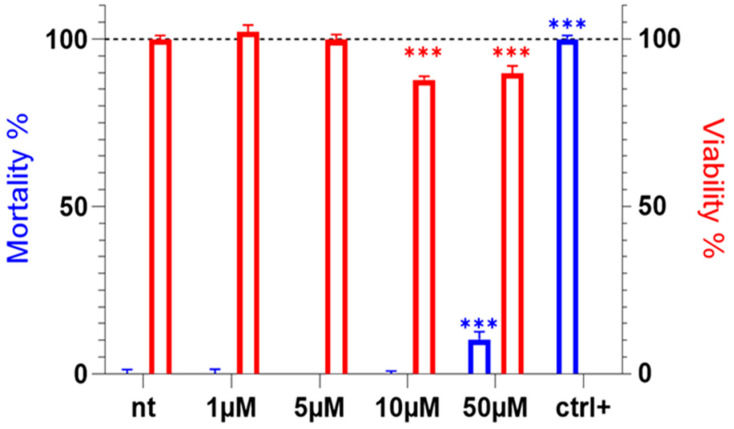
Evaluation of Tb11a cytotoxicity on HEK293T cells. HEK293T were either non-treated (nt) or treated with increasing Tb11a concentrations (from 1 µM to 50 µM). Cell mortality (blue) was normalized to positive control (lysis buffer). Cell viability (red) was normalized to positive control (non-treated). Error bars represent SEM ***: *p* < 0.001 compared to non-treated cells (N = 3).

**Figure 2 toxins-17-00379-f002:**
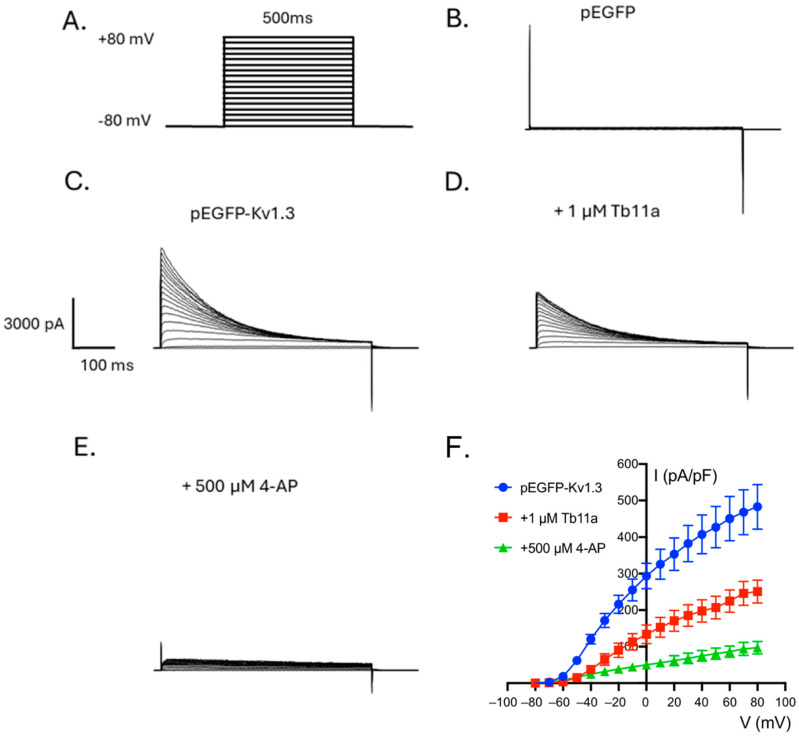
Effect of Tb11a on Kv1.3 currents. (**A**) Stimulation protocol for electrophysiological measurements. (**B**) Representative traces obtained with pEGFP transfected cells. (**C**–**E**) Representative traces obtained of pEGFP-Kv1.3 transfected cells on basal condition (**C**), after 1 µM Tb11a application (**D**) or after addition of 500 µM 4-AP (**E**). (**F**) Current-voltage curves obtained with basal current (blue), after addition of 1 µM Tb11a (red) and 4-AP (green) (Error bars represent SEM *n* = 4).

**Figure 3 toxins-17-00379-f003:**
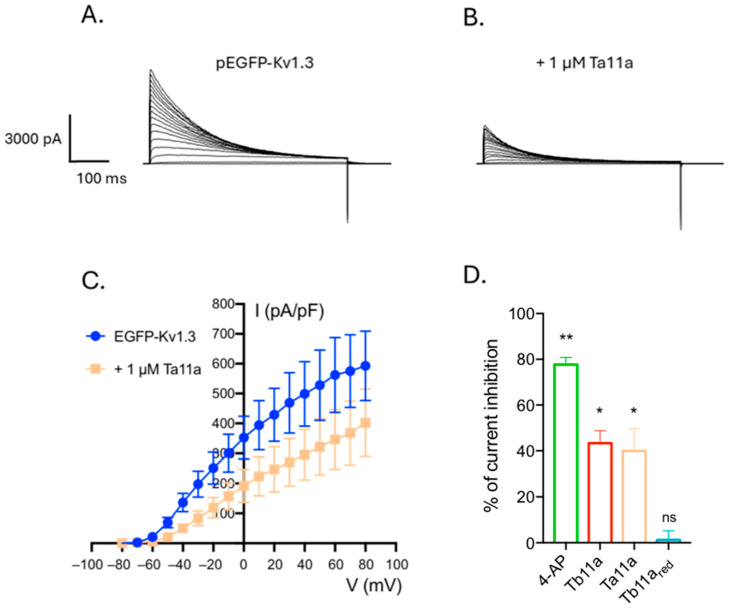
Effect of the Tb11a homolog Ta11a on Kv1.3 current and activity comparison with Tb11a. (**A**,**B**). Representative traces obtained of pEGFP-Kv1.3 transfected cells on basal condition (**A**) and upon 1 µM addition of Ta11a (**B**). (**C**) Current voltage curves obtained before (blue) and after (orange) addition of 1 µM Ta11a. (**D**). Percentage of current inhibition at +40 mV upon treatment with 500 µM 4-AP (green), 1 µM oxidized Tb11a (red), 1 µM Ta11a (orange) and reduced Tb11a (Tb11a_red_, light blue) Error bars represent SEM *n* = 4 or 5. **: *p* < 0.005, *: *p* < 0.05, ns: nonsignificant difference compared to non-treated cells.

**Figure 4 toxins-17-00379-f004:**
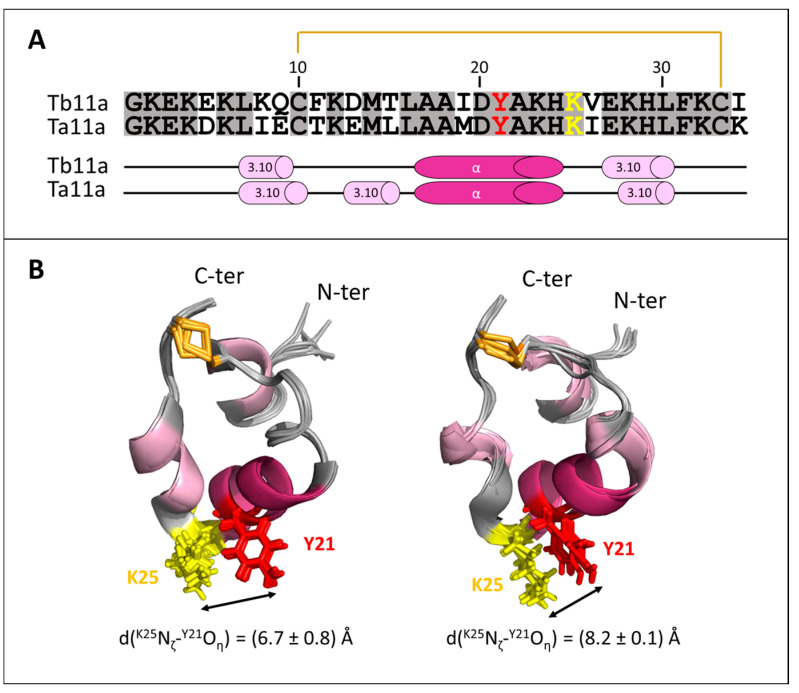
(**A**) Sequence alignment of Tb11a and Ta11a highlighting a few key features. Identical residues are shaded in gray, the C10-C33 disulfide bridge is marked in bright orange, the potential functional dyad residues are depicted in red (Y21) and in yellow (K25), and the secondary structural elements are shown: 3.10-helices in light pink and α-helices in dark pink. (**B**) Structural overlays of the two toxins using the same color-code as in panel (**A**). (**Left**): Overlay of the 15 NMR models of Tb11a (PDB 8PWT). (**Right**): Overlay of the 5 homology models generated for Ta11a.

**Figure 5 toxins-17-00379-f005:**
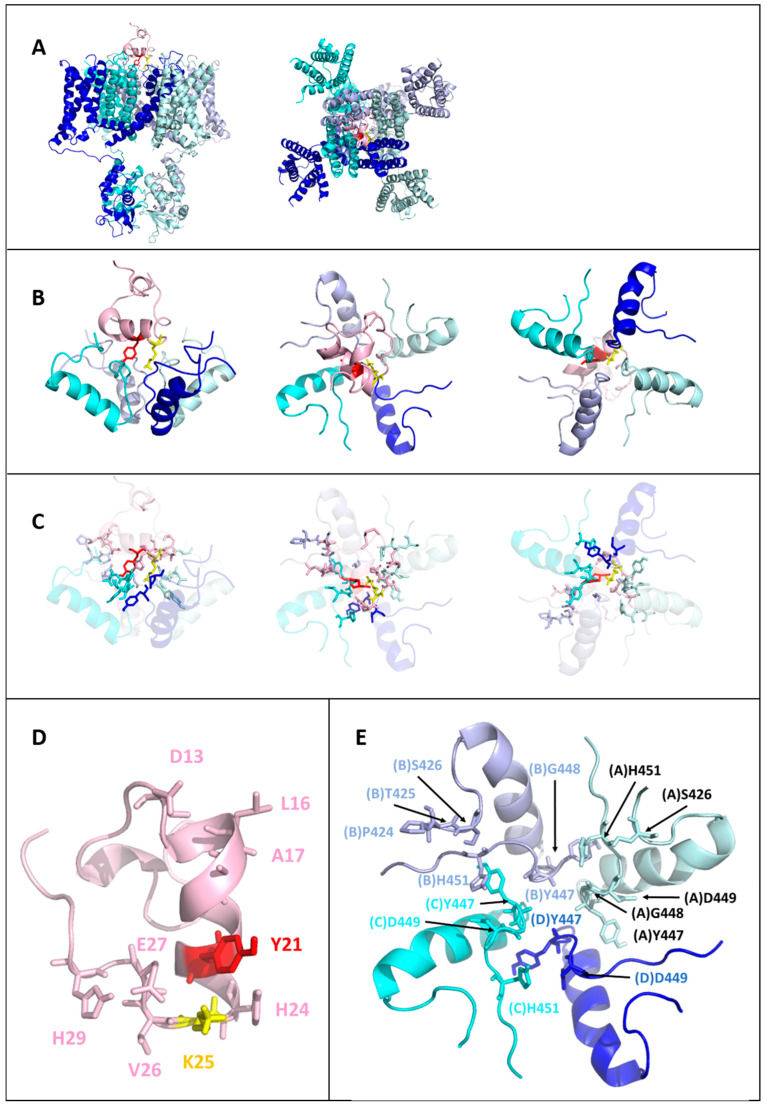
Structural insights into Tb11a binding to Kv1.3: overview of the toxin-channel interaction network. (**A**) Overall view of the Kv1.3 channel (blue to cyan gradient for the four subunits) in complex with Tb11a (pink). The functional dyad residues Y21 (red) and K25 (yellow) are shown as sticks. The two orientations—side view (**left**) and top view (**right**)—highlight the positioning of the toxin relative to the channel. (**B**) Close-up views of Tb11a interacting with Kv1.3, emphasizing the spatial arrangement of the functional dyad residues Y21 and K25. For clarity, only residues 423 to 455 of each Kv1.3 subunit are shown. The left panel shows a side view, the middle panel a top view, and the right panel a bottom view. (**C**) All residues involved in the Tb11a- Kv1.3 interface are represented as sticks. For clarity, helices are shown with high transparency. The views are identical to panel B: side view (**left**), top view (**middle**), and bottom view (**right**). (**D**) Structure of Tb11a with key residues involved in the interaction network represented as sticks. The functional dyad residues, Y21 (red) and K25 (yellow), are prominently displayed. (**E**) Kv1.3 residues involved in the interaction network with Tb11a are highlighted as sticks. Residues from the four subunits (blue to cyan gradient) are labeled for clarity, emphasizing their spatial arrangement and contribution to the interface.

**Figure 6 toxins-17-00379-f006:**
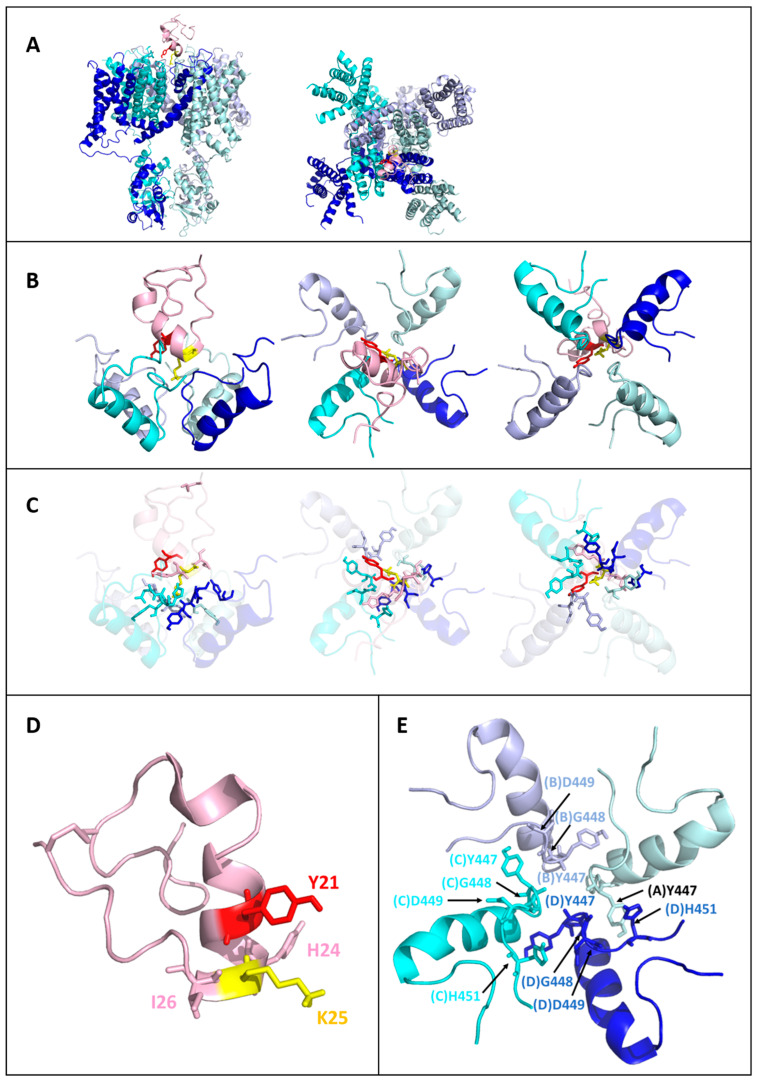
Structural insights into Ta11a binding to Kv1.3: overview of the toxin-channel interaction network. (**A**) Overall view of the Kv1.3 channel (blue to cyan gradient for the four subunits) in complex with Ta11a (pink). The functional dyad residues Y21 (red) and K25 (yellow) are shown as sticks. The two orientations—side view (**left**) and top view (**right**)—highlight the positioning of the toxin relative to the channel. (**B**) Close-up views of Ta11a interacting with Kv1.3, emphasizing the spatial arrangement of the functional dyad residues Y21 and K25. For clarity, only residues 423 to 455 of each Kv1.3 subunit are shown. The left panel shows a side view, the middle panel a top view, and the right panel a bottom view. (**C**) All residues involved at the interface between Ta11a and Kv1.3 are represented as sticks. For clarity, helices are shown with high transparency. The views are identical to panel B: side view (**left**), top view (**middle**), and bottom view (**right**). (**D**) Structure of Ta11a with key residues involved in the interaction network represented as sticks. The functional dyad residues, Y21 (red) and K25 (yellow), are prominently displayed. (**E**) Kv1.3 residues involved in the interaction network with Ta11a are highlighted as sticks. Residues from the four subunits (blue to cyan gradient) are labeled for clarity, emphasizing their spatial arrangement and contribution to the interface.

**Figure 7 toxins-17-00379-f007:**
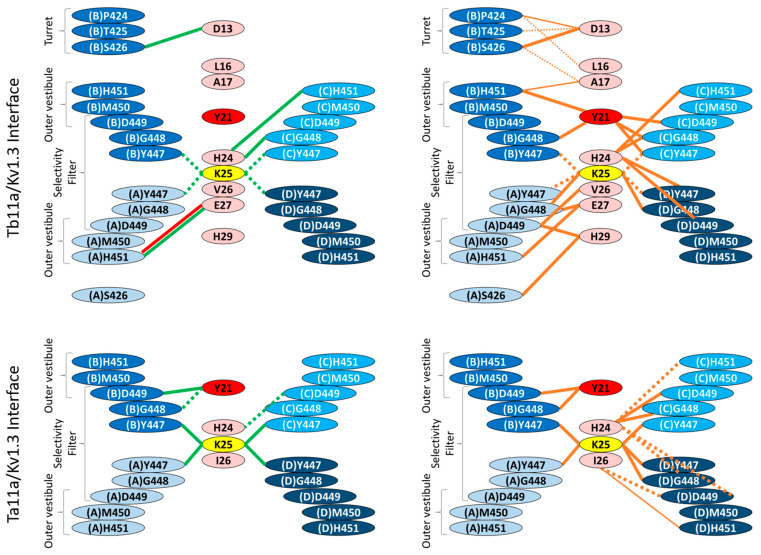
Comparison of the interaction networks of the two toxins Tb11a and Ta11a with the Kv1.3 potassium channel: hydrogen bonds and salt bridges (**left**) vs. hydrophobic interactions (**right**). The residues involved at the interface of the toxins are represented in pink, with the functional dyad residues emphasized in yellow (K25) and red (Y21). The four subunits of the Kv1.3 channel are shown in a blue gradient. On the left, green lines represent hydrogen bonds, and red lines indicate salt bridges. On the right, orange lines depict hydrophobic interactions. On both sides, the thickness of the lines reflects the number of contacts, while the line style indicates prevalence: solid lines represent interactions present in over 50% of the selected structures, and dashed lines indicate interactions occurring in 25% to 50% of structures. This figure highlights the symmetrical and distinct interaction networks of Tb11a and Ta11a with the pore-turret, outer vestibule, and selectivity filter regions of the Kv1.3 channel.

**Table 1 toxins-17-00379-t001:** Comparative summary of the selection process of the Tb11a-Kv1.3 and Ta11a-Kv1.3 conformations generated by docking.

Criteria	Tb11a-Kv1.3	Ta11a-Kv1.3
**None (all refined structures in water)**	200	200
K25 in selectivity filter	80	111
2 H-bond with Y447	63	19
3 H-bonds with Y447	7	10
4 H-bonds with Y447	10	57
**K25 in selectivity filter and conserved structural compactness of the toxin**	42	45
2 H-bond with Y447	34	0
3 H-bonds with Y447	3	0
4 H-bonds with Y447	5	45
**Haddock score**	−61.48 ± 6.28	−36.36 ± 5.44

## Data Availability

The original contributions presented in this study are included in the article/Appendix A. Further inquiries can be directed to the corresponding authors.

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
