# Peer review of "A New Kv1.3 Channel Blocker from the Venom of the Ant Tetramorium bicarinatum"

_toxins, 2025, doi:10.3390/toxins17080379_

Round 1

Reviewer 1 Report

Comments and Suggestions for Authors

Presented manuscript, titled:” A new Kv1.3 channel blocker from the venom of the ant Tetramorium bicarinatum” describes a new characterization of a previously identified venom peptide Tb11a. Tb11a, a 34 amino acid long peptide with one disulfide bridge in its structure, was tested for cytotoxic properties on different human cell lines and for the ability to block potassium channels. Tb11a showed lack of cytotoxic properties, but was found to inhibit, to a different degree, two subtypes of potassium channels. Based on previously published results, the authors evaluated Tb11a and a homologous peptide Ta11a, from Tetramorium africanum species, on voltage-gated Kv1.3 and calcium activated KCa1.1 channels. Tb11a blocked 47.7% and Ta11a 45.8% of the Kv1.3 potassium current at 1 µM. Tb11a blocked KCa1.1 channel by 24.8% at 1 µM. The authors then proceeded to modeling studies of both peptides with the Kv1.3 channel. They focused on the role of functional dyad Y21-K25 found in both peptides, in interaction with the channel. Since NMR structure of Tb11a was previously obtained, Ta11a structure was determined by homology. Both peptides’ structures were used to create 3D models of interaction with Kv1.3 using cryo-EM structure of a Fab-Shk/Kv1.3 complex. The authors collected a set of data suggesting pivotal role of the dyad in the channel modulation as well as indicating disulfide bridge as a crucial element of the peptide’s three-dimensional structure that allows for favorable conformation within the channel.  The results of this study can be further used to created peptide analogs with higher selectivity and potency towards Kv1.3 channel.

Although the reviewer finds this work interesting and well written, there are major and minor corrections the authors should implement to make this study more complete.

Major suggestions:  

  • Please improve citation in the Introduction section of the manuscript (as indicated under minor suggestions)
  • The authors emphasize the role of the dyad in blocking potassium channel through modeling of Tb11a and Ta11a. The important step to validate this hypothesis would be mutating one of the residues, either Y21 or K25, and modeling it to confirm the hypothesis. One step further would be obtaining an analog with one of the residues being substituted by Ala and testing it using electrophysiology. The reviewer strongly suggests doing one of those two experiments.
  • The authors point to results obtained by Peigneur et al, suggesting that the Tb11a peptide they worked with was in a reduced form, therefore it did not block the potassium subtypes it was tested on, but showed cytotoxicity in oocytes. Unfortunately, the authors did not attempt to validate their observation experimentally. Reducing the disulfide bond with DTT or TCEP would allow the authors to test their hypothesis of the disulfide bridge importance using electrophysiology and to test for cytotoxicity, making it an additional control in the human cell lines they tested.
  • The reviewer disagrees with the use of the phrase: potent or strong blockage of Kv1.3 in reference to Tb11a peptide. The potent inhibitors of that subtype have low nM to pM IC50 (a summary of Kv1.3 inhibitors can be found in a review by Cheng et al. (2024) titled: “Voltage-gated potassium channel 1.3: A promising molecular target in multiple disease therapy”,) while Tb11a blocks only 47% of the peak current at 1 µM, suggesting low µM IC50.
  • In the discussion section, the authors refer to A4 superfamily of peptides, a Figure (possibly included in the supporting information) with some representative sequences would be usefully to the reader.

Minor suggestions:

Page 1, line 11: The authors should rewrite the sentence starting with: “Tb11a did show no or low cytotoxicity…” to emphasize that the result is a part of their own new research, and not a part of the previous study. It is a bit confusing in the current form.

Page 1, paragraph between lines 35-42: please provide a reference

Page 2, paragraph between lines 43-45: please provide a reference

Page 2, line 54: please provide a reference for a “functional dyad”

Page 2, line 72: please provide a reference for the sentence: “For example, the potent Kv blocker ShK from sea anemone features a K22-57 Y23 dyad that demonstrates this mechanism.”

Page 2, line 58: please provide a reference for the following statement: “However, some toxins, despite possessing this dyad, behave as if they do not, relying instead on multi-site stabilization to block the channel.”

Page 3, line 63-66: please provide a reference

Page 3, line 78: please replace “Additionally” with an equivalent word like “Moreover”, since the word additionally was use at the beginning of a previous sentence.

Page 4, line 142: In the supporting information, the authors should provide HPLC chromatogram of the peptide along with the MS result.

Page 4, line 177: please replace PyMO with PyMOL

Page 4: remove lines 181-183

Page 5, line 185: please provide a reference.

Page 5, line 197-199: please provide (in parenthesis), which cell lines were used as a representation of epithelial breast cells and which as neuronal. It would be great if the authors explained why those cell lines not others were used for this experiment. In addition, Figure S1 does not include a control – this needs to be changed.

Page 5, line 204: Standard Error of the Mean should be capitalized without the full stops as SEM. It also needs to be corrected in line 251, as well as in legend to Figures S1 and S2 of the supporting information

Page 5, line 217: please provide a full name 4-aminopyridne followed by an abbreviation in parenthesis.

Page 5/6, line 217-220: please simplify the sentence starting with: “To further investigate Tb11a’s activity….”  

Page 6/7, lines 233-238: In the section starting with “Tb11a displays an original structure….” the authors state that Tb11a contains two putative functional dyads. Please identify both dyads.

Page 13, line 387: please provide a reference for either the sentence starting with: “The precise position-…” or in line 390 for “This common feature is notably found….

Page 16, line 485 the word myrmecinea should be capitalize and italicized

Page 16, line 507- 508 a full stop is needed at the end of the sentence: “Therefore, further characterization of the properties of Tb11a and Ta11a will be needed.”

Paxillin is introduced in Figure S2 and Figure S2 legend, but no information about it is provided anywhere is the manuscript. This should be addressed. Is it paxillin or paxillin AP?

Reviewer 2 Report

Comments and Suggestions for Authors

The Authors studied the effect of the fully structured and dissulfide-bonded Tb11a peptide on mammalian cells. Firstly, they studied cytotoxicity of the peptides and showed lack of cytotoxicity at concentrations up to 10 mikormols. Then, they discovered that both peptides aaplied at 1 mikromol concentration, are blockers of Kv1.3 channels in HEK293T cells and, to a lesser extent, K(Ca)1.1 channels. Results of electrophysiological studies were followed by structural analysis of the peptides. Models of the peptide-channel complexes were shown as well as toxins/channel interfaces. The paper also contains discussion, conclusions and reference list containing 29 positions.

The article is interesting and worthy for publication after a small revision. Three points need to be addressed:

1) Are the inhibitory effects on the channels reversible ?

2) The Authors should perform patch-clamp studies applying more concentrations of the peptides and build a dose-respnse curves for both channels.

3) The Authors should also study putative inhibitory effects of the peptides on other types of Kv1.x channels, such as, for example, Kv1.1 channels.

Round 2

Reviewer 1 Report

Comments and Suggestions for Authors

The authors responded to all the suggestions satisfactory.

There is one more corrections they should make: 

Description to Figure 3, Line 178: "and reduced Tb11a (Tb11ared, purple)" - but the Figure shows the reduced form in light blue. Either the text or the color in the figure for this analog should be changed. 

Author Response

Description to Figure 3, Line 178: "and reduced Tb11a (Tb11ared, purple)" - but the Figure shows the reduced form in light blue. Either the text or the color in the figure for this analog should be changed. 
Thank you for the comment.

The text of the legend was changed in “light blue”. The text of the supplemental figure S2 was changed too.

In addition, the word myrmicinae was corrected page 15 line 422